# K^+^ Channel Tetramerization Domain 5 (KCTD5) Protein Regulates Cell Migration, Focal Adhesion Dynamics and Spreading through Modulation of Ca^2+^ Signaling and Rac1 Activity

**DOI:** 10.3390/cells9102273

**Published:** 2020-10-12

**Authors:** Jimena Canales, Pablo Cruz, Nicolás Díaz, Denise Riquelme, Elías Leiva-Salcedo, Oscar Cerda

**Affiliations:** 1Program of Cellular and Molecular Biology, Institute of Biomedical Sciences (ICBM), Faculty of Medicine, Universidad de Chile, 8380453 Santiago, Chile; jimenacanales@med.uchile.cl (J.C.); pcruznunez@gmail.com (P.C.); nicolasdiazvl@gmail.com (N.D.); 2Millennium Nucleus of Ion Channel-Associated Diseases (MiNICAD), Universidad de Chile, 8380453 Santiago, Chile; 3Department of Biology, Faculty of Chemistry and Biology, Universidad de Santiago de Chile, 9170022 Santiago, Chile; denise.riquelme@usach.cl (D.R.); elias.leiva@usach.cl (E.L.-S.); 4The Wound Repair Treatment and Healing (WoRTH) Initiative, 8380453 Santiago, Chile

**Keywords:** KCTD5 protein, cell migration, focal adhesions, spreading, Ca^2+^ signaling, Rac1

## Abstract

Cell migration is critical for several physiological and pathophysiological processes. It depends on the coordinated action of kinases, phosphatases, Rho-GTPases proteins, and Ca^2+^ signaling. Interestingly, ubiquitination events have emerged as regulatory elements of migration. Thus, the role of proteins involved in ubiquitination processes could be relevant to a complete understanding of pro-migratory mechanisms. KCTD5 is a member of Potassium Channel Tetramerization Domain (KCTD) proteins that have been proposed as a putative adaptor for Cullin3-E3 ubiquitin ligase and a novel regulatory protein of TRPM4 channels. Here, we study whether KCTD5 participates in cell migration-associated mechanisms, such as focal adhesion dynamics and cellular spreading. Our results show that KCTD5 CRISPR/Cas9- and shRNA-based depletion in B16-F10 cells promoted an increase in cell migration and cell spreading, and a decrease in the focal adhesion area, consistent with an increased focal adhesion disassembly rate. The expression of a dominant-negative mutant of Rho-GTPases Rac1 precluded the KCTD5 depletion-induced increase in cell spreading. Additionally, KCTD5 silencing decreased the serum-induced Ca^2+^ response, and the reversion of this with ionomycin abolished the KCTD5 knockdown-induced decrease in focal adhesion size. Together, these data suggest that KCTD5 acts as a regulator of cell migration by modulating cell spreading and focal adhesion dynamics through Rac1 activity and Ca^2+^ signaling, respectively.

## 1. Introduction

Cell migration is a fundamental process involved in a plethora of physiological and pathophysiological events, such as embryonic development, immune response, wound healing and metastasis of cancer cells [1,2,3,4]. This process consists of cycles of five consecutive steps: cell polarization, membrane extension at the leading edge, adhesion to cell substratum, generation of traction forces and detachment of rear edge [5]. All those events are the result of the spatial–temporal coordination of several molecular signals, where Rho-GTPases and Ca^2+^ play a central role [6]. Rho-GTPases are associated with the regulation of cell membrane extension, adhesion to the extracellular matrix and generation of contractile forces during migration. Cdc42 and Rac1 are linked to filopodia and lamellipodia formation at the leading edge, respectively, while RhoA is responsible for the maturation of focal adhesions and formation of stress fibers [7,8,9,10]. On the other hand, Ca^2+^ is involved in the molecular mechanisms leading to the formation of nascent focal adhesions in the front of migrating cells and in those leading to the focal adhesion disassembly in the rear [11,12,13,14,15,16]. In addition, Ca^2+^ participates in the regulation of actin cytoskeleton dynamics through several downstream effectors, promoting the dynamic protrusions and retraction in the front of migrating cell, and mediating the contractile forces in the rear and cell body [11,17,18,19]. In accordance with the similarity of Ca^2+^ and Rho GTPases actions in migration-related mechanisms, it has been described that Ca^2+^ regulates the activity of RhoA and Rac1 in some cellular models [20,21].

The Potassium Channel Tetramerization Domain Containing (KCTD) is a family of small soluble proteins, characterized by sharing sequence similarity with the T1 domain of the voltage-gated K^+^ channels (Kv channels) [22]. Reports suggest that most of the KCTD members may interact with the ubiquitin ligases (E3) type proteins, Cullin Ring Ligases (CRL) [22]. Indeed, it has been demonstrated that the most studied CRL, Cullin 3 (Cul3), interacts with KCTD2, KCTD5, KCTD6, KCTD9, KCTD11, KCTD13, KCTD17, KCTD21, and TNFAIP1 through their Bric-a-Brac, Tramtrack, Broad complex (BTB) domain [23,24,25,26,27,28,29,30]. In these complexes, KCTD proteins can participate as adaptors for the interaction between Cul3 and its substrates, regulating the selectivity of this E3 ubiquitin ligase for specific target of ubiquitination. Interestingly, some members of the KCTD family participate in mechanisms involved in cytoskeleton rearrangement and cell migration. KCTD13 (BACURD1) and TNFAIP1 (BACURD2) regulate fiber stress formation and promote cell migration by allowing the Cul3-dependent RhoA ubiquitination and subsequent proteasomal degradation [24]. Conversely, KCTD10 mediates RhoB ubiquitination and lysosomal degradation, which is linked to loss of contractility in endothelial cells and to an increase in Rac1 activity in breast cancer cells [31,32].

KCTD5 is the first member of the KCTD family whose tridimensional structure has been described [33]. Furthermore, its interaction with Cul3 has been extensively studied [23,34,35,36,37]. KCTD5 participates in such uneven processes as anti-proliferative response, *Helicobacter pylori* adherence to gastric cells and sleep regulation [37,38,39]. Interestingly, KCTD5 precludes the activation of the known pro-migratory and pro-carcinogenic AKT pathway by switch-off the G-protein coupled receptors (GPCR) signaling through the Gβγ subunit proteasomal degradation [36]. Additionally, KCTD5 associates with TRPM4 channels and regulates its Ca^2+^ sensitivity [40]. Conversely, KCTD5 is overexpressed in breast cancer samples and is involved in the regulation of cell migration in a TRPM4-dependent manner [40]. All these recent reports elucidate a plethora of cellular responses associated with KCTD5. However, the molecular mechanisms underlying these effects remain unclear. Here, we provide novel data demonstrating that KCTD5 regulates cell migration, spreading, focal adhesion dynamics, Rac1 activity and intracellular Ca^2+^ levels of melanoma cells. Thus, we propose a novel role for KCTD5 as a negative modulator of cell migration-associated processes by regulating Rac1 activity and Ca^2+^ response.

## 2. Materials and Methods

### 2.1. Reagents

RPMI 1640 Medium (catalog #31800022), Trypsin-EDTA (catalog #15400054), Penicillin-Streptomycin (catalog #15140122), Puromycin (catalog #A1113803), Lipofectamine^TM^ LTX with PLUS reagent (catalog #15338030), Fura-2 AM (catalog #F1201), Alexa Fluor^TM^ 488 Phalloidin (catalog #A12379), Alexa Fluor^TM^ 555 Phalloidin (catalog #A34055), Hoechst 33258 reagent (catalog #H3569), Pierce^TM^ BCA Protein Assay kit (catalog #23225), SuperSignal^TM^ West Pico PLUS Chemiluminescent Substrate (catalog #34580), TRIzol^TM^ reagent (catalog #15596026), Ribolock RNase Inhibitor (catalog #EO0381), RevertAid Reverse Transcriptase (catalog #EP0441) and Ionomycin (catalog #I24222) were obtained from Thermo Fisher Scientific (Carlsbad, CA, USA). Fetal Bovine Serum (FBS) (catalog #SV30160.03) was from GE Healthcare Life Science (Chicago, IL, USA). Sodium chloride (NaCl) (catalog #1064045000) was obtained from Merck (Darmstadt, Germany). Paraformaldehyde (catalog #158127), Sucrose (catalog #S0389), Human plasma fibronectin (catalog #FC010), IGEPAL^®^ CA-630 (NP-40) (catalog #I3021), Tween^®^ 20 (catalog #P1379), Triton X-100 (catalog #10789704001), Tris base ULTROL^®^ Grade (catalog #648311), Sodium orthovanadate (catalog #567540), Phenylmethylsulfonyl fluoride (PMSF) (catalog #78830) and Sodium fluoride (NaF) (catalog #S7920) were obtained from MilliporeSigma (Burlington, MA, USA). Protease Inhibitor Cocktail (catalog #M221) was obtained from VWR Life Science AMRESCO (West Chester, PA, USA). SensiMix SYBR Hi-ROX kit (catalog #QT605-05) was acquired in Bioline (London, UK).

### 2.2. Plasmids

Plasmid encoding Enhanced Green Fluorescent Protein (EGFP)-fused-KCTD5 protein (EGFP-KCTD5) was previously described [38]. pEGFP-C1 plasmid was obtained from Clontech Laboratories (Mountain View, CA, USA). KCTD5 human shRNA plasmid kits (catalog #TL303786) were obtained from Origene (Rockville, MD, USA). Plasmids encoding myc-tagged Rac1-Q61L and Rac1-T17N (pRK5-myc-Rac1-Q61L and pRK5-myc-Rac1-T17N) were a gift from Gary Bokoch (Addgene plasmids #12983 and #12984, respectively). KCTD5 Double Nickase plasmid (m) (catalog #sc-427353-NIC) and Control Double Nickase plasmid (catalog #sc-437281) were obtained from Santa Cruz Biotechnology (Dallas, TX, USA).

### 2.3. Antibodies

Antibodies used for immunofluorescence and immunoblot experiments are listed in Table 1.

### 2.4. Cell Culture and Transfection

B16-F10 cells and B16-F10-derived cell lines were culture in RPMI 1640 medium supplemented with 10% *v*/*v* Fetal Bovine Serum and antibiotics (10,000 U/mL penicillin, 10 ug/mL streptomycin). Cells were incubated at 37 °C in humidified atmosphere and 5% CO_2_. For KCTD5 knockdown, B16-F10 cells were transiently transfected with plasmid encoding a shRNA^Scramble^ sequence as a control or with plasmids encoding two different shRNA sequences against KCTD5 (shRNA^KCTD5^ #1 and shRNA^KCTD5^ #2). These plasmids also encode EGFP protein, which allowed us to identify transfected cells in experiments involving image analyses. The KCTD5-knockdown assays were performed 72 h post-transfection. For KCTD5 overexpression, B16-F10 cells were transiently transfected with EGFP-fused KCTD5 encoding plasmid (pEGFP-C1-KCTD5) previously generated as described [38]. pEGFP-C1 plasmid was used as a control. KCTD5 overexpression experiments were performed 48 h after transfection. For Rac1 mutants overexpression, B16-F10 cells were transiently transfected with myc-Rac1-Q61L or myc-Rac1-T17N encoding plasmids, and an empty vector was used as a control. These experiments were carried out 48 h after transfection. All the transfections were performed using Lipofectamine LTX^TM^ reagent and following the instructions provided by the manufacturer.

### 2.5. KCTD5 Knockout B16-F10 (B16-F10^kctd5-/-^) and Control B16-F10 (B16-F10^Control^) Cells Generation

B16-F10 cells were transfected with Double Nickase mouse KCTD5 plasmid or with Double Nickase Control plasmid using Lipofectamine LTX^TM^ according to the manufacturer’s instructions. Seventy-two hours post-transfection, cells were grown in a selection medium containing 2 µg/mL puromycin for two weeks. Selected cells were expanded and KCTD5 knockout was validated by immunoblot and qPCR as described [40].

### 2.6. Immunoblot Analyses

Cells were washed with cold Dulbecco’s phosphate buffered saline (DPBS, pH 7.4) and lysed with cold lysis buffer (150 mM NaCl, 50 mM Tris-HCl (pH 7.4), 0.5% *v*/*v* NP-40, 5 mM NaF, 1 mM PMSF, 1 mM sodium orthovanadate and 1X Protease Inhibitor Cocktail) for 20 min at 4 °C. The lysates were centrifuged at 13,000× *g* at 4 °C. Proteins were quantified with Pierce^TM^ BCA protein kit according to the manufacturer’s instructions. Thirty to fifty micrograms of total proteins were separated in 12% polyacrylamide gels and transferred to a nitrocellulose membrane. Blots were blocked with 0.1% *v*/*v* Tween-20, 4% *w*/*v* non-fat milk in Tris-buffered saline (TBS: 50 mM Tris, pH 7.5, 150 mM NaCl) and then were probed with primary antibodies. Primary antibodies were detected with appropriate horseradish peroxidase-conjugated secondary antibodies. Secondary antibody was detected with SuperSignal^TM^ chemiluminescent substrate. Images were acquired with a miniHD9 (UVITEC, Cambridge, UK) chemiluminiscence photodocumentation system using NineAlliance software (UVITEC, Cambridge, UK). Protein levels were quantified by scanning densitometric analysis using UnScan it v6.1 software (Silk Scientific, Inc., Orem, UT, USA). Values obtained for proteins of interest were normalized to α-tubulin levels.

### 2.7. mRNA Expression Analyses by RT-qPCR

mRNA from B16-F10^Control^ and B16-F10*^kctd5-^*^/*-*^ cells was isolated using TRIzol^TM^ Reagent according to the manufacturer’s instructions. cDNA was obtained as described [40]. Quantitative PCR reactions were prepared using SensiMix SYBR Hi-ROX kit following supplier’s instructions. The primers used for KCTD5 and β-actin cDNAs amplification were: KCTD5: Forward: 5′-GGAGCTGCTGGGATTCCTTT-3′, Reverse: 5′-GTCAGTCTGCACAGTACCCC-3′; β-actin: Forward: 5′-TGACGTGGACATCCGCAAAG-3′; Reverse: 5′-CTGGAAGGTGGACAGCGAGG-3’. Reactions were carried out using StepOne Real-Time Step System (Applied Biosystems, Foster City, CA, USA) as indicated [40]. Relative expression was determined using the double delta Ct method. KCTD5 mRNA levels were normalized to β-actin mRNA.

### 2.8. Boyden Chamber Transwell Migration Assays

Transwell migration assays were performed as described [41]. Briefly, cells were serum-starved 16 h before assay. After that, 6.0 × 10^4^ cells per condition were seeded in Transwell chambers (Corning Costar Corp., MA, USA. Catalog #CLS3422) and migration was stimulated by adding 10% *v*/*v* FBS in the lower chamber for 18 h at 37 °C. Non-migrating cells were removed and migrating cells were fixed and stained with 0.2% *w*/*v* violet crystal/10% *v*/*v* ethanol. Migrating cells were counted and expressed as fold over control.

### 2.9. Spreading Assay

Spreading assays were performed as described [41]. Briefly, cells were serum-starved 16 h prior to assay. Then, 3.0 × 10^4^ cells were seeded in 5 µg/mL fibronectin-coated 12 mm coverslips using a medium containing 10% *v*/*v* FBS. Forty-five or ninety minutes post-seeding, cells were fixed in fixative solution (4% *w*/*v* formaldehyde—freshly prepared from paraformaldehyde—4% *w*/*v* sucrose in DPBS, pH 7.4) for 15 min at 4 ºC. Then, cells were permeabilized and blocked with blocking solution (4% *w*/*v* nonfat dry milk/0.1% *v*/*v* Triton X-100 in DPBS) for 45 min at room temperature. Actin cytoskeleton was detected with Alexa-555 or Alexa-488-conjugated phalloidin. Nuclei were detected with Hoechst 33258 reagent. Images were acquired with Olympus Disk Scanning Unit confocal microscope (Olympus, Tokyo, Japan) and a 20× objective or a 60× oil objective. Cells showing sheet-like membrane protrusions with branched F-actin were considered as lamellipodial-like cells as described [41,42]

### 2.10. Focal Adhesion Analyses

Focal adhesion analyses were performed as described [41,43]. Briefly, cells were seeded in 5 µg/mL fibronectin-coated 12 mm coverslips for 16 h. After that, cells were serum-starved for 4 h and then stimulated with 10% *v*/*v* FBS for 30 min. Next, cells were fixed, permeabilized and blocked as described above. Focal adhesions were recognized with mouse anti-vinculin monoclonal antibody, which was detected with the Alexa555-conjugated anti-mouse antibody. Nuclei were detected with Hoechst 33258 reagent. Images were acquired with Olympus Disk Scanning Unit confocal microscope (Olympus, Tokyo, Japan) and a 60× oil objective.

### 2.11. Focal Adhesions Dynamics Analyses

For focal adhesion dynamics measurements, cells were transfected with plasmid encoding mCherry-paxillin construct and 48 h post-transfection were seeded in 5 µg/mL fibronectin-coated 25 mm coverslips. Then, cells were serum-starved for 16 h and focal adhesion assembly/disassembly was promoted by adding 10% *v*/*v* FBS. Images were acquired in a recording chamber in an Olympus Disk Scanning Unit confocal microscope (Olympus, Tokyo, Japan) every 30 s for 30 min [41,43]. Analyses were performed using Focal Adhesion Analysis Server [44] complemented by manual analyses of ln(I/I_0_) vs. time as described [45].

### 2.12. Intracellular Calcium Measurements

Cells were starved 4 h before the experiment. Next, cells were incubated with 5 µM Fura-2 probe in Krebs solution (140 mM NaCl, 2.5 mM KCl, 2 mM CaCl_2_ (2.5 mM CaCl_2_ for KCTD5-knockdown experiments), 1 mM MgCl_2_, 10 mM HEPES, 10 mM glucose, pH 7.4) for 30 min at 37 °C. After that, cells were washed with Krebs solution for 15 min at 37 °C. Then, coverslips were mounted and maintained in Krebs in a recording chamber in an Eclipse Ti2-U inverted microscope (Nikon, Tokyo, Japan) controlled with Micromanager 1.4 software (Vale Lab, University of California, San Francisco, CA, USA). Images from Fura-2 were acquired every 20 s by alternate excitation at 340 and 380 nm and emissions were captured at 510 nm. The ratio of 340 nm fluorescence to 380 nm fluorescence was measured and all data were represented as a change in ratio units ((F − F_0_)/F_0_).

### 2.13. Statistical Analyses

Data shown are the mean ± SD (or mean ± SEM in Figure 5) of at least three independent experiments. Data were analyzed using two-tailed unpaired Student’s *t*-test to compare two conditions. For multiple comparisons, One-way or Two-way Analysis Of Variance (ANOVA) tests were used, as appropriate. Comparisons between groups were performed using Sidak’s post-test to compare two groups and Tukey’s post-test to compare three or more groups. Analyses were carried out using GraphPad Prism v8.0 (GraphPad Prism, San Diego, CA, USA).

## 3. Results

### 3.1. KCTD5 Depletion Increases Cell Migration of B16-F10 Cells

To study whether KCTD5 is involved in migration, we used the highly migratory model of B16-F10 melanoma cells. Cells were transiently knocked down for the KCTD5 expression employing shRNA strategy (Figure 1A) and cell migration was evaluated. We observed that KCTD5 depletion, obtained through two different shRNA constructs, promoted a two-fold increase in the average of cell migration compared to the respective control cells (Figure 1B). A similar trend, although at a lesser extent, was observed in the MCF-7 breast cancer cell line (Appendix A). Alternatively, we generated stable KCTD5 Knockout B16-F10 (B16-F10*^kctd5-^*^/*-*^) and Control B16-F10 (B16-F10^Control^) cell lines based in CRISPR/Cas-9 system (Figure 1C,D). Consistent with the effects observed in KCTD5-knocked down B16-F10 cells, B16-F10*^kctd5-^*^/*-*^ increased on average three times the cell migration compared to B16-F10^Control^ (Figure 1E). We did not find changes in cell viability in KCTD5-depleted B16-F10 cells during migration assays (Appendix A), suggesting that the effect observed in migration is not due to cell death and/or proliferation effects. We then overexpressed KCTD5-EGFP in B16-F10^Control^ and B16-F10*^kctd5-^*^/*-*^ cells to corroborate the impact of KCTD5 in cell migration (Figure 1F). As expected, KCTD5-EGFP expression completely precluded the KCTD5 depletion-enhanced cell migration (Figure 1G). Thus, these results indicate that KCTD5 negatively regulates the cell migration of B16-F10 cells.

### 3.2. KCTD5 Depletion Enhances the Spreading of B16-F10 Cells

Cell migration is highly dependent on the dynamic interaction between the cells and the extracellular matrix. This interaction through focal complexes and focal adhesions, as well as the coordinated actin cytoskeleton rearrangement, are essential for effective cell mobility [6]. We then evaluated whether KCTD5 affects these parameters. We performed spreading assays to determine whether KCTD5 affects early adhesion and cytoskeleton dynamics. The mean area of KCTD5-knockdown B16-F10 cells seeded in fibronectin-coated coverslips was 1.8-fold higher than the mean area of control cells at 45 min post-seeding, and 1.5-fold (shRNA^KCTD5^ #1) and 1.8-fold (shRNA^KCTD5^ #2) higher than control at 90 min of spreading (Figure 2A,B). Similarly, B16-F10*^kctd5-/-^* cells display in average 1.6-fold and 1.4-fold increase in the area of the cells seeded in fibronectin-coated coverslips at 45 and 90 min, respectively, compared to their respective control cells (Figure 2D,E). On the other hand, the percentage of lamellipodial-like cells at 45 min of spreading was variably increased by shRNA^KCTD5^ (38.6 ± 12.2% for control cells, 50.0 ± 10.6% and 69.1 ± 5.6% for shRNA^KCTD5^#1- and shRNA^KCTD5^#2-transfected cells, respectively). These proportions were akin at 90 min post-seeding (39.8 ± 3.2% for control cells, 46.1 ± 8.2% and 62.8 ± 5.9% for shRNA^KCTD5^#1- and shRNA^KCTD5^#2-transfected cells, respectively) (Figure 2C). Consistent with these results, the percentage of lamellipodial-like cells at 45 min post-seeding increased from 18.0 ± 8.8% for B16-F10^Control^ cells to 40.7 ± 10.2% for B16-F10*^kctd5-/-^*, but this difference was not observed at 90 min (18.8 ± 9.1% for B16-F10^Control^ cells vs. 28.9 ± 5.1% for B16-F10*^kctd5-/-^*) (Figure 2F). Following the proposed role of KCTD5 in cell spreading, the expression of EGFP-KCTD5 completely reverted the increased cell area and percentage of lamellipodial-like cells promoted by KCTD5 depletion (Figure 2G–I). Together, these findings indicate that KCTD5 also regulates the spreading of B16-F10 cells.

### 3.3. Cell Spreading and Migration Promoted by KCTD5-Depletion Is Dependent on Rac1 Activity

Rac1 is a member of the Rho-GTPase family involved in early adhesion and lamellipodia formation during cell spreading and migration [46]. We then evaluated whether Rac1 activity is implicated in the higher cell spreading linked to KCTD5 depletion. For this, B16-F10^Control^ and B16-F10*^kctd5-/-^* cells were transfected with constructs encoding two Rac1 variants: Rac1-Q61L (constitutively active) and Rac1-T17N (dominant-negative) (Figure 3D). Constitutively active Rac1 mutant increased twice the mean spreading area of control cells, with no effect on B16F10*^kctd5-/-^* cells (Figure 3A,B). Conversely, the dominant-negative Rac1 mutant did not affect the cell area of control, but it decreased the area of B16-F10*^kctd5-/-^* cells to a similar level as B16-F10^Control^ (Figure 3A,B). Consistently, constitutively active Rac1 increased the percentage of lamellipodial cells in B16-F10^Control^ condition from 24.4 ± 9.2% to 70.9±3.8%, without affecting this parameter in B16-F10*^kctd5-/-^* cells (72.2 ± 11.5% and 72.6 ± 9.7% for Mock- and Rac1-Q61L-transfected B16-F10*^kctd5-/-^* cells, respectively) (Figure 3C). Moreover, the dominant-negative Rac1 mutant decreased the percentage of lamellipodial-like cells in B16-F10*^kctd5-/-^* to 32.2 ± 8.3%, but it did not have a significant impact in B16-F10^Control^ cells (24.4 ± 9.2% for Mock and 34.6 ± 2.3% for Rac1-T17N) (Figure 3C). All these results suggest that KCTD5 regulates the cell spreading of B16-F10 cells through modulation of Rac1 activity.

Given the effects promoted by constitutively active Rac1 mutant in cell spreading, we next evaluated whether these results are associated with cell migration. The Rac1-Q61L mutant showed a trend toward increasing the migration of B16-F10^Control^ cells, although this difference was not significant (*p* = 0.0555). Consistent with the response observed on spreading of B16-F10*^kctd5-/-^* cells expressing Rac-Q61L, constitutively active Rac1 did not affect the migration of KCTD5-depleted cells (Figure 3E). This suggests that KCTD5 could exert its effects on the migration of B16-F10 cells by regulating the Rac1 activity.

### 3.4. KCTD5 Depletion Affects Focal Adhesion Dynamics of B16-F10 Cells

Once the migrating cells adhere to the substratum, the adhesion structures must be coordinately assembled at the leading edge and disassembled at the rear for an effective displacement [6]. To determine whether KCTD5 is involved in these dynamics, we first analyzed the effect of this protein on the size and number of focal adhesions in completely adhered cells. We found that KCTD5-silenced B16-F10 cells showed smaller focal adhesions compared to the respective control cells, such as the average area of focal adhesions per cell was 2.9 ± 0.7 μm^2^ for control and 1.9 ± 0.5 μm^2^ for shRNA^KCTD5^-transfected cells (Figure 4A,B). However, the number of these structures was not affected by KCTD5 silencing (Figure 4C). Additionally, the body shape of KCTD5-knockdown B16-F10 cells was less elongated than control cells (Figure 4A), an effect possibly associated with the lower cell tension derived from smaller focal adhesions. In accordance with the effect of KCTD5 silencing on focal adhesion size, the average focal adhesion area of B16-F10*^kctd5-^*^/*-*^ cells was lesser than the focal adhesion average area of B16-F10^Control^ cells (2.3 ± 0.5 μm^2^ and 2.7 ± 0.7 μm^2^, respectively) (Figure 4D,E). Moreover, B16-F10*^kctd5-^*^/*-*^ cells did not display any impact on the number of focal adhesions per cell (Figure 4F). Consistent with the lower size of focal adhesions, B16-F10*^kctd5-^*^/*-*^ cells showed less distinguishable stress fibers, as was evidenced by F-actin staining (Figure 4D).

We then measured the focal adhesion dynamics in living cells using the mCherry-Paxillin construct as a focal adhesion sensor (Appendix A, Figure 4G). We observed that focal adhesion disassembly was more frequent than focal adhesion assembly (Figure 4H,I), but we did not find significant differences in the number of these events between control and KCTD5-knockdown conditions (disassembling focal adhesions: 39 ± 28 and 68 ± 31; assembling focal adhesions: 12 ± 12 and 3 ± 1 for shRNA^Scramble^ and shRNA^KCTD5^, respectively). Interestingly, we found that KCTD5-knockdown enhanced ~25% of the focal adhesion disassembly rate compared to control cells (Figure 4G,I), in accordance with the reduced focal adhesion size in KCTD5-depleted B16-F10 cells. Furthermore, the KCTD5 silencing did not have any impact on focal adhesion assembly rates (Figure 4G,H). Overall, these observations suggest that the KCTD5 depletion decreases focal adhesion size presumably by increasing the disassembly rate of these structures.

### 3.5. KCTD5 Depletion Impacts in Serum-Induced Ca^2+^ Signaling

Ca^2+^ is a second messenger key in several cellular processes, including cell migration [47]. Specifically, Ca^2+^ participates in mechanisms linked to focal adhesion dynamics and actin cytoskeleton rearrangement [11,14,48]. Conversely, KCTD5 is a regulatory protein of TRPM4, a well-known ion channel involved in Ca^2+^ signaling regulation [40,49]. We then studied whether KCTD5 has a role in the regulation of Ca^2+^ response to serum stimulus. Ca^2+^ response was diminished in KCTD5-knockdown B16-F10 cells, as well as the maximal peak observed in these cells was ~65% lower than that in control cells (Figure 5A,B). The kinetics of intracellular Ca^2+^ changes also presented differences, such as KCTD5-silenced B16-F10 cells exhibited a slope of Ca^2+^ increase slightly lower and a Ca^2+^ decrease slower than control cells (Figure 5A). Conversely, B16-F10^Control^ cells expressing EGFP-KCTD5 showed an enhanced serum-induced Ca^2+^ response and a 2-fold increased maximal peak compared to B16-F10^Control^ cells expressing EGFP, although both conditions showed a similar kinetic for Ca^2+^ changes (Figure 5C,D). Moreover, the decreased serum-induced Ca^2+^ signal observed for EGFP-transfected B16-F10*^kctd5-^*^/*-*^ cells compared to their respective control was rescued when KCTD5 expression was re-established by EGFP-KCTD5 (Figure 5E). Consequently, the decrease of ~37% in the maximal peak exhibited by EGFP-transfected B16-F10*^kctd5-^*^/*-*^ compared to EGFP-transfected B16-F10^Control^ cells was completely reverted in the B16-F10*^kctd5-^*^/*-*^ cells expressing exogenous EGFP-KCTD5 (Figure 5F). Consistent with kinetics observed for KCTD5-knockdown condition, the slope of Ca^2+^ decrease in EGFP-expressing B16-F10*^kctd5-^*^/*-*^ cells was lower than EGFP-transfected B16-F10^Control^ and EGFP-KCTD5-transfected B16-F10*^kctd5-^*^/*-*^ cells (Figure 5E). These results indicate that KCTD5 expression levels could exert a modulatory role in the serum-induced Ca^2+^ signaling.

### 3.6. Serum-Induced Ca^2+^ Rise Is Necessary for KCTD5-Mediated Focal Adhesion Regulation

Since KCTD5 depletion decreased the serum-induced Ca^2+^ rise in B16-F10 cells, we evaluated whether this observation is related to the KCTD5 depletion effects in late cell adhesion. Thus, we reverted the KCTD5 depletion-induced Ca^2+^ signal decrease using the ionophore ionomycin. We observed that ionomycin precluded the KCTD5 knockdown-induced focal adhesion area diminution in a Ca^2+^-containing medium (1.6 ± 0.4 μm^2^ in DMSO condition versus 2.2 ± 0.7 μm^2^ with ionomycin) (Figure 6A,B). Intriguingly, KCTD5 depletion increased the number of focal adhesions per cell compared to control cells in the DMSO condition (75 ± 34 focal adhesions per cell for control versus 97 ± 37 focal adhesions per cells for KCTD5-knockdown) (Figure 6C). This difference was not observed in the presence of ionomycin, suggesting a nonspecific effect of the vehicle (Figure 6C). To assure that the effects of ionomycin on focal adhesion size were certainly a result of Ca^2+^ entry, we performed the experiment in a Ca^2+^-free medium to avoid the massive entry of Ca^2+^ through the ionophore. We observed that ionomycin in a Ca^2+^-free medium was unable to revert the KCTD5 depletion-induced reduction of focal adhesion size, such as the average focal adhesion area in this condition (1.5 ± 0.4 μm^2^) (Figure 6D,E) was similar to that observed in 2 mM Ca^2+^ medium containing DMSO (1.6 ± 0.4 μm^2^) (Figure 6A,B). Notably, ionomycin in a Ca^2+^-free medium induced an apparent lower number of focal adhesions per cell (55 ± 26 focal adhesions per cell for control and 62 ± 30 focal adhesions per cell for KCTD5 depletion) (Figure 6F). Control cells seemed more sensitive to this effect, given that the number of focal adhesions increased significantly to 92 ± 51 focal adhesions per cell by ionomycin-induced Ca^2+^ entry (Figure 6F). In summary, the ionomycin-promoted Ca^2+^ entry was sufficient to prevent the decrease in adhesion structures area induced by KCTD5 depletion, suggesting that KCTD5 could modulate the focal adhesion size by regulating the serum-induced Ca^2+^ signaling.

## 4. Discussion

Cell migration is a process dependent on coordinated spatiotemporal signals involving Rho-GTPases activity and Ca^2+^ signaling [6]. These mechanisms promote the assembly/disassembly of focal complexes and/or focal adhesions, as well as the rearrangement of the actin cytoskeleton, necessary for an effective cell movement [7,11,14,17,50]. Recently, post-translational mechanisms have been linked to cell migration. Thus, ubiquitination has been associated with this cellular response, mainly through proteasome-dependent protein degradation [51]. Consistent with this idea, different proteins participating in ubiquitination machinery have been linked to cell migration-related processes [51].

Some members of the KCTD superfamily have been proposed as adaptors for the E3-ubiquitin ligase Cul3 [23,24,25,26,27,28,29]. In this study, we demonstrated that KCTD5 influences cell migration, cell spreading, and focal complexes and focal adhesion dynamics in a model of murine melanoma cells. Here, KCTD5 acts as a negative regulator, as the KCTD5 depletion increases all those parameters. Other members of KCTD proteins—KCTD13 (BACURD1) and TNFAIP1 (BACURD2)—in association with Cul3, have the opposite effect, increasing cell migration through degradation of RhoA [24]. Similarly, the KCTD10/Cul3 complex has been associated with RhoB degradation, promoting F-actin arrangement, consistent with an increased Rac1 activity [31,32]. Since KCTD proteins confer selectivity for Cul3 substrates, it is plausible that different adaptors for this E3 Ubiquitin–ligase complex lead to different cellular responses. In fact, Cul3 promotes or inhibits cell migration depending on specific substrate adaptors participating in the complex and, consequently, on specific ubiquitination targets [24,52,53,54,55]. Our results support this idea, given the contrasting cell response observed for KCTD5 compared to other KCTD proteins. Therefore, our findings could contribute to the understanding of the complexity of the Cul3 role in cell migration. On the other hand, KCTD5 has been shown to be related to TRPM4-promoted migration in mouse fibroblasts (MEFs) [40]. Additionally, in vivo KCTD5 morpholino-based attenuation suggests that KCTD5 promotes cell migration in zebrafish [40]. Conversely, KCTD5 is a negative regulator of the pro-migratory AKT pathway in other cell lines [36]. Thus, KCTD5 role in cell migration-related mechanisms may vary depending on particular interacting-partners and/or cell context. Given those questions, it is a challenge for future research to determine whether the effects here reported for KCTD5 are dependent on its role as a Cul3 adapter and, in that case, the identity of the molecular targets that could explain the results observed.

As an approach to this question, we evaluated the role of Rac1 due to its function in the formation of lamellipodia during early adhesion and early steps of migration [7,8]. The obtained results suggest that Rac1 is involved in the higher migration and cell spreading linked to KCTD5 depletion in melanoma cells, such as a constitutively active Rac1 construct enhanced both cellular responses in control, but not in B16-F10*^kctd5-^*^/*-*^ cells. Furthermore, the expression of a dominant-negative Rac1 mutant prevented the cell spreading in B16-F10*^kctd5-^*^/*-*^, but not in control cells, strengthening the idea that Rac1 mediates the effects induced by KCTD5 depletion. Although regulation of migration has been associated with Rac1 ubiquitination and subsequent proteasomal degradation in other cell models [56,57,58,59], we found that neither Rac1 levels were affected by KCTD5 depletion nor proteasome inhibition promoted Rac1 accumulation, independently of KCTD5 presence (Appendix A). Thus, our data suggest that KCTD5 does not participate directly in Rac1 degradation, but it could regulate the Rac1 activity. Accordingly, KCTD5 might indirectly regulate Rac1 activity through RhoA activity modulation, as RhoA promotes an inhibitory effect on Rac1 activity [46,60], and Cul3 in association with some KCTD family members are linked to RhoA stability [24]. Thus, a possible role of KCTD5 in the control of Rac1 activity through modulation of RhoA activity and/or stability should be considered for further studies. Moreover, RhoA and Rac1 activity can be modulated by Guanine nucleotide exchange factors (GEFs, activators) or GTPase activating proteins (GAPs, inactivators) [60]. Thus, KCTD5 might also target these proteins to modulate the activity of Rac1. Consistently, Cul3 and BTB proteins KBTB6/7 have been associated with Rac1 activity regulation by promoting the degradation of Rac1-activator Tiam1 [53]. If KCTD5 performs a similar action or if its presumably effect on Rac1 activity is independent of ubiquitination-mediated mechanisms is an open question to answer in future research.

We demonstrated that the serum-induced Ca^2+^ signal was decreased by KCTD5 depletion in B16-F10 cells. This suggests that another possible molecular identity that could be targeted by KCTD5 are ion channels. Indeed, other KCTD proteins regulate ion channels function [40,61,62]. Interestingly, KCTD5 is involved in the regulation of TRPM4 activity, a well-described ion channel related to cell migration [40,41,63,64,65]. Therefore, it could be plausible that the KCTD5-dependent effects in Ca^2+^ response are the result of the action of KCTD5 on the ion channels function.

We observed that KCTD5 depletion also decreased the size of focal adhesions without affecting the number of focal adhesions per cell. Although both parameters represent the cell’s ability to adhere to the extracellular matrix, focal adhesion size only can predict the cell speed of migrating cells [66]. Thus, the decreased focal adhesion size observed in KCTD5-depleted cells is consistent with the resulting higher migration in the same condition. Moreover, the decrease in focal adhesion area promoted by KCTD5 depletion was congruent with the reported enhancement of focal adhesion turnover rate, suggesting that the faster focal adhesion disassembly is responsible for the lower size of adhesion structures observed in KCTD5-depleted cells.

The focal adhesion disassembly process is dependent on high Ca^2+^ concentrations [14,16], but the spatiotemporal regulation of Ca^2+^ concentration changes are also relevant for this event occurs [11,15,50]. Here, we reported a global decreased intracellular Ca^2+^ signaling in KCTD5-depleted cells. Furthermore, ionomycin-dependent Ca^2+^ entry recovered the effects of KCTD5 silencing on the size of focal adhesions, although the Ca^2+^ output from endoplasmic reticulum promoted by ionomycin in a Ca^2+^ free medium was not sufficient to revert these differences. These data suggest that the effects of KCTD5 on focal adhesion turnover are at least partially dependent on its effect on Ca^2+^ signaling. Moreover, they suggest that endoplasmic reticulum Ca^2+^ is altered in KCTD5-depleted cells, which could explain in part the lower serum-induced Ca^2+^ signal exhibited by these cells. Further studies are required to determine the contribution of KCTD5 to the Ca^2+^ dynamics in migrating cells to dissect with more precision the mechanisms involved in the KCTD5-mediated focal adhesion turnover.

Together, our data indicate that KCTD5 is a negative regulator for the migration of melanoma cells by affecting two crucial molecular players of this process: Rac1 activity and Ca^2+^ signaling (Figure 7). If these effects are dependent on the KCTD5 role as a substrate adaptor for the E3-ubiquitin ligase Cul3 is an interesting issue to explore in future research. Furthermore, the contrasting pro-migratory effect of KCTD5 observed in fibroblasts and zebrafish [40] compared to the KCTD5 negative effect on migration of melanoma cells and breast cancer cells (Appendix A) here reported, raises the additional question about how the cellular context impacts in the KCTD5 role on cell migration-related mechanisms. Considering the cell models used here, it emerges the idea about the relevance of KCTD5 in cancer. Several members of the KCTD protein family are linked to different types of cancer [30,32,40,67,68,69,70,71,72,73,74]. Moreover, migratory features acquisition is crucial for malignant cells spread from localized tumors to distant organs [75]. Thus, our findings could help to elucidate new regulatory mechanisms involving KCTD5 as an inhibitor of processes associated with cancer metastasis.

## Figures and Tables

**Figure 1 cells-09-02273-f001:**
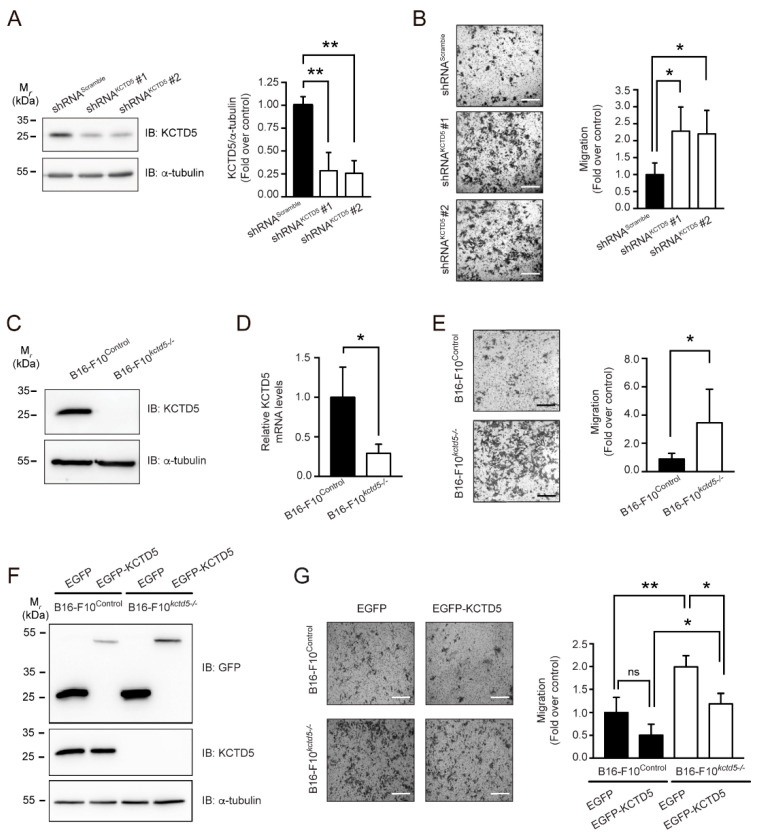
Potassium Channel Tetramerization Domain 5 (KCTD5) regulates migration of B16-F10 cells. (**A**) Evaluation of KCTD5 levels by immunoblot in B16-F10 cells transfected with shRNA^Scramble^, shRNA^KCTD5^# 1 or shRNA^KCTD5^ #2 encoding plasmids. The KCTD5 levels were normalized to α-tubulin loading control. Graph represents the relative KCTD5 levels for each condition (mean ± SD; *n* = 3; One-way ANOVA followed by Tukey’s multiple comparisons test, ** *p* < 0.01). (**B**) Transwell Boyden chamber migration assays of B16-F10 cells transfected with shRNA^Scramble^, shRNA^KCTD5^ #1 or shRNA^KCTD5^ #2 encoding plasmids. Cells were stimulated with 10% *v*/*v* serum for 18 h. Scale bar = 500 μm. Graph represents the relative migration for each condition (mean ± SD; *n* = 5; One-way ANOVA followed Tukey’s multiple comparisons test, * *p* < 0.05). (**C**) Representative immunoblots showing KCTD5 levels in B16-F10^Control^ and CRISPR/Cas9-based KCTD5 Knockout B16-F10 cells (B16-F10*^kctd5-^*^/*-*^). α-tubulin was used as loading control. (**D**) Analysis of KCTD5 mRNA expression by RT-qPCR. The KCTD5 mRNA levels were normalized to β-Actin mRNA. Graph represents the relative KCTD5 mRNA levels for B16-F10^Control^ and B16-F10*^kctd5-^*^/*-*^ cells (mean ± SD; *n* = 3; two-tailed unpaired Student’s *t*-test, * *p* < 0.05). (**E**) Transwell Boyden chamber migration assays of B16-F10^Control^ and B16-F10*^kctd5-^*^/*-*^ cells. Cells were stimulated with 10% *v*/*v* serum for 18 h. Scale bar = 500 μm. Graph represents the relative migration for each condition (mean ± SD; *n* = 3; two-tailed unpaired Student’s *t*-test, * *p* < 0.05). (**F**) Representative immunoblots showing levels of EGFP, EGFP-KCTD5 and endogenous KCTD5 in EGFP- or EGFP-KCTD5-transfected B16-F10^Control^ and B16-F10*^kctd5-^*^/*-*^ cells. α-tubulin was used as loading control. (**G**) Transwell Boyden chamber migration assays of B16-F10^Control^ and B16-F10*^kctd5-^*^/*-*^ cells transfected with EGFP or EGFP-KCTD5 encoding plasmids. Cells were stimulated with 10% *v*/*v* serum for 18 h. Scale bar = 500 μm. Graph represents the relative migration for each condition (mean ± SD; *n* = 3; Two-way ANOVA followed by Sidak’s multiple comparisons test, ns = not significant, * *p* < 0.05, ** *p* < 0.01).

**Figure 2 cells-09-02273-f002:**
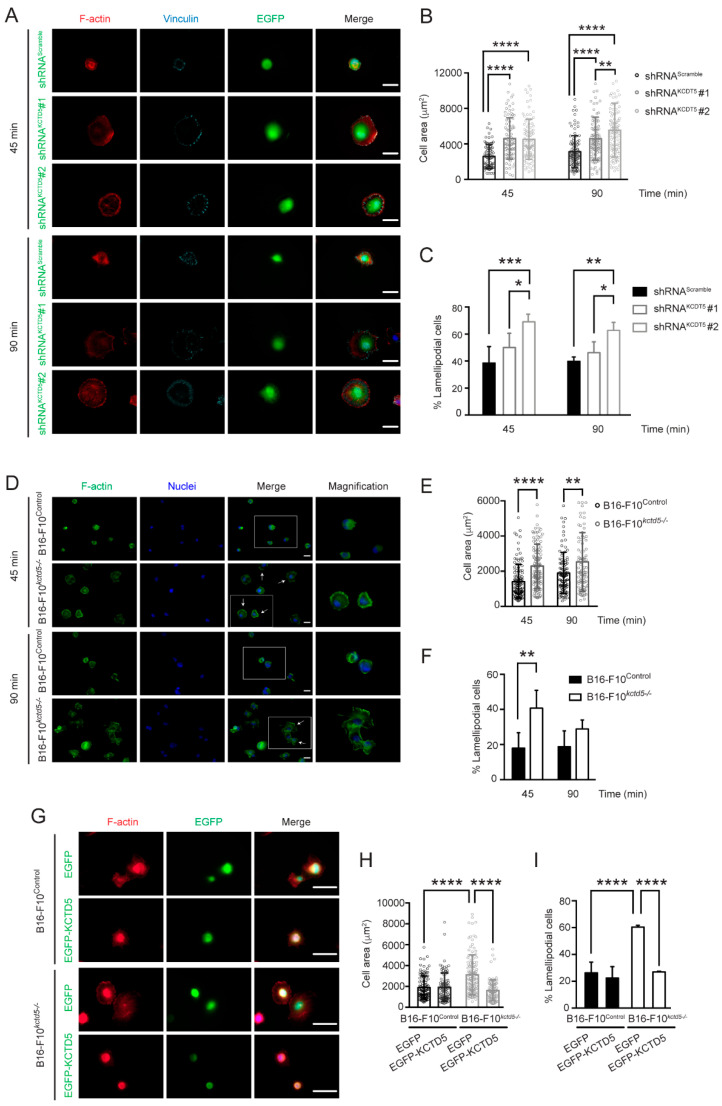
KCTD5 regulates spreading of B16-F10 cells. (**A**) Representative images of shRNA^Scramble^-, shRNA^KCTD5^ #1- or shRNA^KCTD5^ #2-transfected B16-F10 cells incubated for 45 or 90 min on fibronectin-coated coverslips. Cells were fixed and stained for F-actin with Alexa-555 phalloidin (red). Focal adhesions were labeled with mouse mAb anti-vinculin (cyan). EGFP positive cells were analyzed. Scale bar = 0.25 µm. (**B**) Graph representing the cell area (µm^2^) for each condition (mean ± SD; *n* = 4; Two-way ANOVA followed by Tukey’s and Sidak’s multiple comparisons tests, ** *p* < 0.01, **** *p* < 0.0001). (**C**) Graph representing the percentage of lamellipodial-like cells for each condition (mean ± SD; *n* = 4; Two-way ANOVA followed by Tukey’s and Sidak’s multiple comparisons tests, * *p* < 0.05, ** *p* < 0.01, *** *p* < 0.001). (**D**) Representative images of B16-F10^Control^ and B16-F10*^kctd5-/-^* cells incubated for 45 or 90 min on fibronectin-coated coverslips. Cells were fixed and stained for F-actin with Alexa-488 phalloidin (green) and for nuclei with Hoechst reagent (blue). Arrows show lamellipodial-like cells. White boxes show the magnification area. Scale bar = 0.20 µm. (**E**) Graph representing the cell area (µm^2^) for each condition (mean ± SD; *n* = 4; Two-way ANOVA followed by Sidak’s multiple comparisons test, ** *p* < 0.01, **** *p* < 0.0001). (**F**) Graph representing the percentage of lamellipodial-like cells for each condition (mean ± SD; *n* = 4; Two-way ANOVA followed by Sidak’s multiple comparisons test, ** *p* < 0.01). (**G**) Representative images of EGFP- or EGFP-KCTD5-transfected B16-F10^Control^ and B16-F10*^kctd5-/-^* cells incubated for 45 min on fibronectin-coated coverslips. Cells were fixed and stained for F-actin with Alexa-555 phalloidin (red). EGFP positive cells were analyzed. Scale bar = 0.25 µm. (**H**) Graph representing the cell area (µm^2^) for each condition (mean ± SD; *n* = 4; Two-way ANOVA followed by Sidak’s multiple comparisons test, **** *p* < 0.0001). (**I**) Graph representing the percentage of lamellipodial-like cells for each condition (mean ± SD; *n* = 4; Two-way ANOVA followed by Sidak’s multiple comparisons test, **** *p* < 0.001).

**Figure 3 cells-09-02273-f003:**
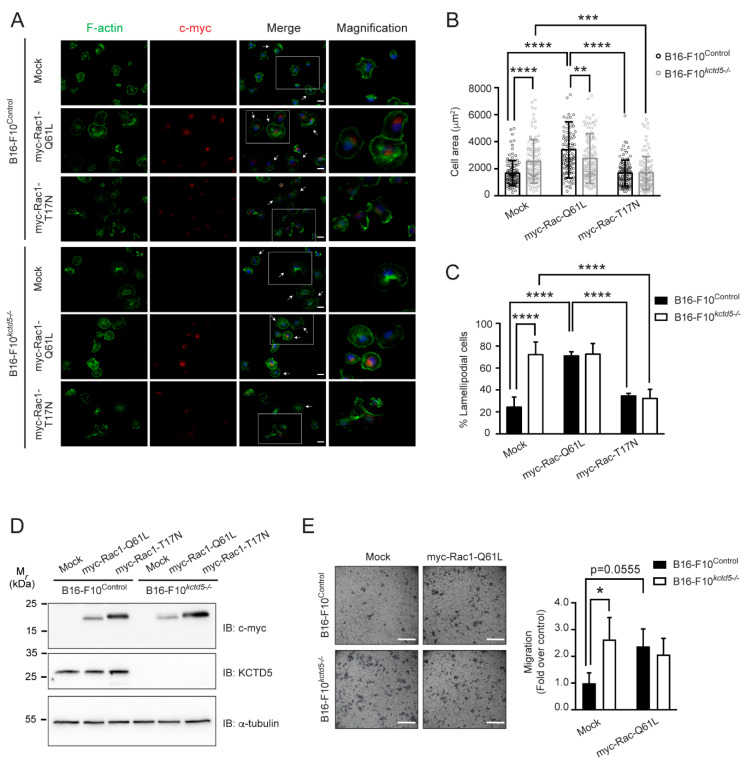
KCTD5 regulates cell spreading and migration through Rac1. (**A**) Representative images of B16-F10^Control^ and B16-F10*^kctd5-^*^/*-*^ cells transfected with empty vector (Mock), myc-Rac1-Q61L (constitutively active) or myc-Rac1-T17N (dominant-negative) encoding plasmids. Cells were seeded on fibronectin-coated coverslips for 45 min, fixed and stained for F-actin with Alexa-488 phalloidin (green). Rac1 mutants-transfected cells were labeled with mouse mAb anti-c-myc (red). Arrows show lamellipodial-like cells. White boxes show magnification area. Scale bar = 0.20 µm. (**B**) Graph representing the cell area (µm^2^) for each condition (mean ± SD; *n* = 4; Two-way ANOVA followed by Tukey’s and Sidak’s multiple comparisons tests, ** *p* < 0.01, *** *p* < 0.001, **** *p* < 0.0001). (**C**) Graph representing the percentage of lamellipodial-like cells for each condition (mean ± SD; *n* = 4; Two-way ANOVA followed by Tukey’s and Sidak’s multiple comparisons tests, **** *p* < 0.0001). (**D**) Representative immunoblot showing levels of myc-Rac1-Q61L (constitutively active), myc-Rac1-T17N (dominant-negative) and KCTD5 in B16-F10^Control^ and B16-F10*^kctd5-^*^/*-*^ cells transfected with empty vector (Mock), myc-Rac1-Q61L or myc-Rac1-T17N encoding plasmids. α-tubulin was used as loading control. (**E**) Transwell Boyden chamber migration assays of B16-F10^Control^ and B16-F10*^kctd5-^*^/*-*^ cells transfected with empty vector (Mock) or myc-Rac1-Q61L encoding plasmid. Cells were stimulated with 10 % *v*/*v* serum for 18 h. Scale bar = 500 µm. Graph represents the relative migration for each condition (mean ± SD; *n* = 3; Two-way ANOVA followed by Sidak’s multiple comparisons tests, * *p* < 0.05).

**Figure 4 cells-09-02273-f004:**
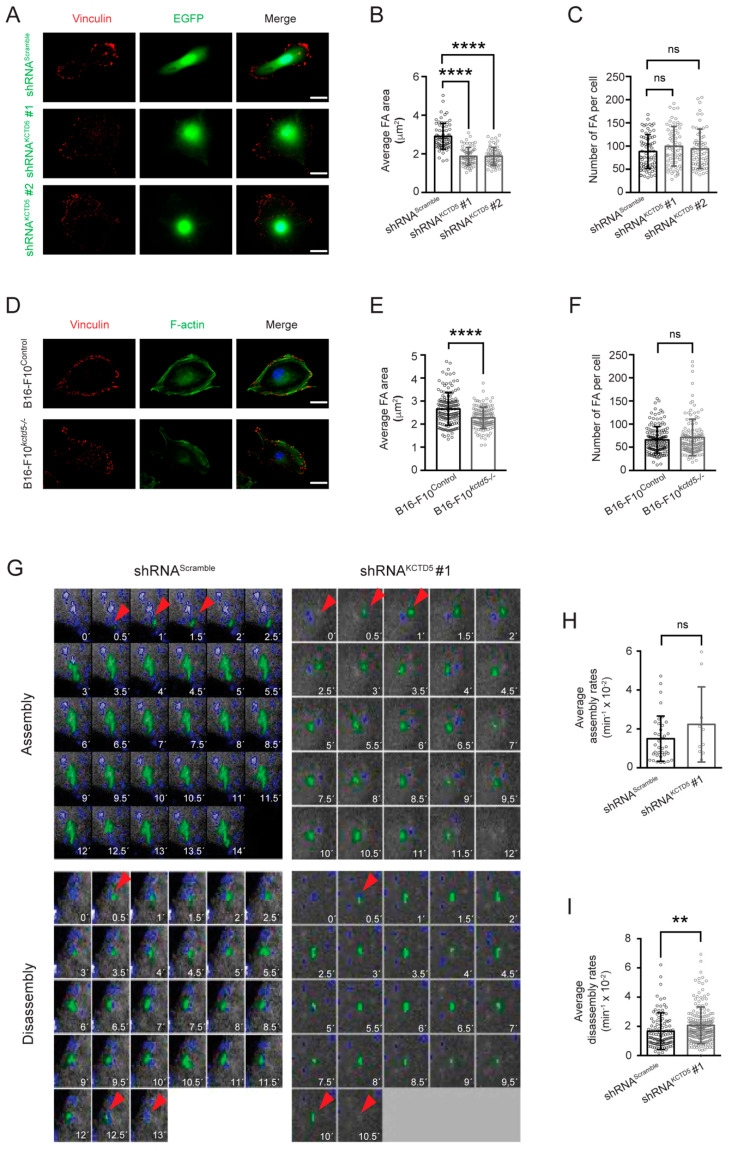
KCTD5 regulates focal adhesion size of B16-F10 cells. (**A**) Representative images of B16-F10 cells transfected with shRNA^Scramble^, shRNA^KCTD5^ #1 or shRNA^KCTD5^ #2 encoding plasmids. Focal adhesions were labeled with mouse mAb anti-vinculin (red). Focal adhesions of EGFP positive cells were analyzed. Scale bar = 25 μm. (**B**) Graph representing the Average of focal adhesion area per cell (µm^2^) for each condition (mean ± SD; *n* = 5; One-way ANOVA followed by Tukey’s multiple comparisons test, **** *p* < 0.0001). (**C**) Graph representing the Number of focal adhesions per cell for each condition (mean ± SD; *n* = 5; One-way ANOVA followed by Tukey’s multiple comparisons test, ns = not significant). (**D**) Representative images of B16-F10^Control^ and CRISPR/Cas9-based KCTD5 Knockout (B16-F10*^kctd5-^*^/*-*^) cells immunostained against the focal adhesion protein vinculin (red). F-actin was visualized by Alexa 488-phalloidin stain. Scale bar = 25 μm. (**E**) Graph representing the Average of focal adhesion area per cell (µm^2^) for each condition (mean ± SD; *n* = 5; two-tailed unpaired Student’s *t*-test, **** *p* < 0.0001). (**F**) Graph representing the number of focal adhesions per cell for each condition (mean ± SD; *n* = 5; two-tailed unpaired Student’s *t*-test, ns = not significant). (**G**) Representative images showing tracked focal adhesions (red arrowhead and green mark) and time (in minutes) from B16-F10 cells co-transfected with mCherry-paxillin construct and shRNA^Scramble^ or shRNA^KCTD5^ #1 encoding plasmids. Serum-starved cells were stimulated with 10% *v*/*v* Fetal Bovine Serum (FBS) to induce the focal adhesion assembly/disassembly. Focal adhesions were tracked by mCherry-paxillin fluorescence. Focal adhesion dynamics were obtained by live-cell time-lapse recording. (**H**) Graph representing the average assembly rate (mean ± SD; *n* = 3; two-tailed unpaired Student’s *t*-test, ns = not significant). (**I**) Graph representing the average disassembly rate (mean ± SD; *n* = 3; two-tailed unpaired Student’s *t*-test, ** *p* < 0.01).

**Figure 5 cells-09-02273-f005:**
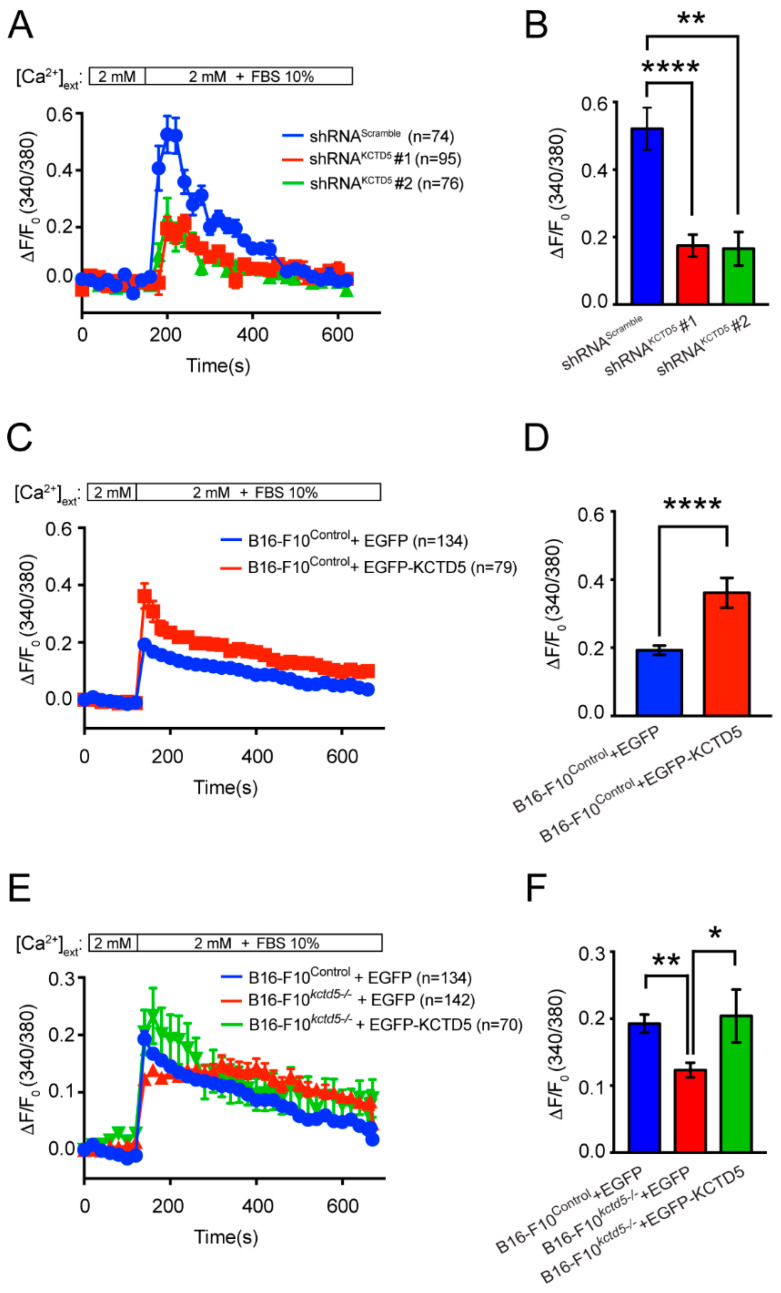
KCTD5 promotes serum-induced Ca^2+^ signals in B16-F10 cells. (**A**) Time courses for normalized fluorescence in B16-F10 cells transfected with shRNA^Scramble^, shRNA^KCTD5^ #1 or shRNA^KCTD5^ #2 encoding plasmids. Cells were loaded with 5 µM Fura-2-AM probe and Ca^2+^ peak was induced by 10% *v*/*v* FBS in 2.5 mM CaCl_2_ Krebs medium. Each point corresponds to the mean ± SEM. (**B**) Graph representing the mean ± SEM of ΔF_maximum_/F_0_ for each condition (*n* = 7; One-way ANOVA followed by Tukey’s multiple comparisons test, ** *p* < 0.01, **** *p* < 0.0001). (**C**) Time courses for normalized fluorescence in B16-F10^Control^ cells transfected with EGFP or EGFP-KCTD5 encoding plasmids. Cells were loaded with 5 µM Fura-2-AM probe and Ca^2+^ peak was induced by 10% *v*/*v* FBS in 2 mM CaCl_2_ Krebs medium. Each point corresponds to the mean ± SEM. (**D**) Graph representing the mean ± SEM of ΔF_maximum_/F_0_ for each condition (*n* = 5; two-tailed unpaired Student’s *t*-test, **** *p* < 0.0001). (**E**) Time courses for normalized fluorescence in B16-F10^Control^ cells expressing EGFP and B16-F10*^kctd5-^*^/*-*^ cells expressing EGFP or EGFP-KCTD5 construct. Cells were loaded with 5 µM Fura-2-AM probe and Ca^2+^ peak was induced by 10% *v*/*v* FBS in 2 mM CaCl_2_ Krebs medium. Each point corresponds to the mean ± SEM. (**F**) Graph representing the mean ± SEM of ΔF_maximum_/F_0_ for each condition (*n* = 5; One-way ANOVA followed by Tukey’s multiple comparisons test, * *p* < 0.05, ** *p* < 0.01).

**Figure 6 cells-09-02273-f006:**
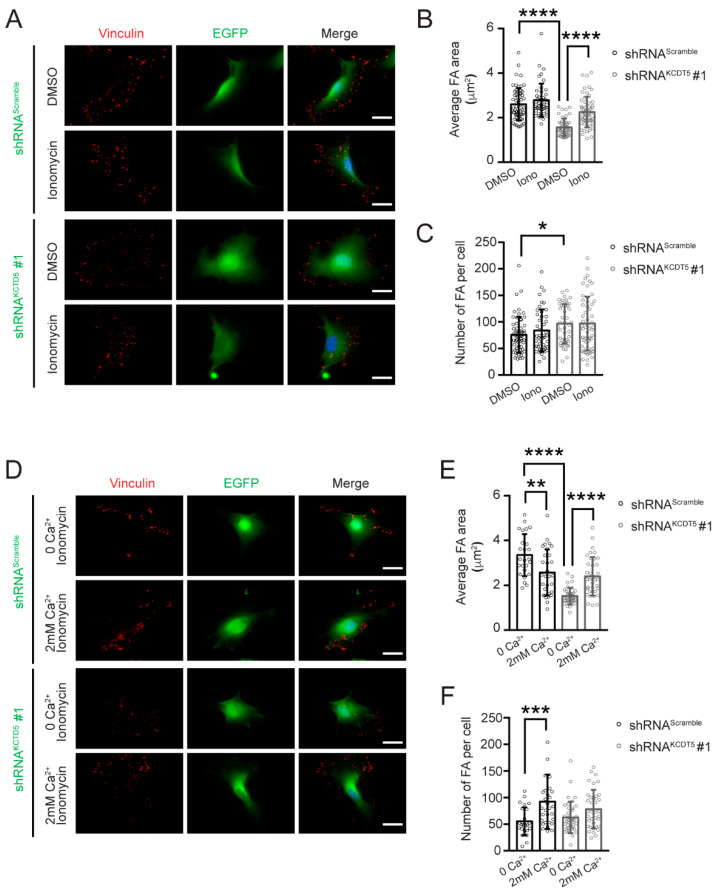
Ca^2+^ mediates the KCTD5-promoted focal adhesion size regulation. (**A**) Representative images of B16-F10 transfected with shRNA^Scramble^ or shRNA^KCTD5^ #1 encoding plasmids. Cells were serum-starved for 4 h and then were treated with 1 μM ionomycin in a 2 mM CaCl_2_ Krebs medium for 20 min. DMSO was used as a vehicle control. Focal adhesions were labeled with mouse mAb anti-vinculin (red). Focal adhesions of EGFP positive cells were analyzed. Scale bar = 25 μm. (**B**) Graph representing the Average of focal adhesion area per cell (µm^2^) for each condition (mean ± SD; *n* = 4; Two-way ANOVA followed by Sidak’s multiple comparisons test, **** *p* < 0.0001). (**C**) Graph representing the Number of focal adhesions per cell for each condition (mean ± SD; *n* = 5; Two-way ANOVA followed by Sidak’s multiple comparisons test, * *p* < 0.05). (**D**) Representative images of B16-F10 transfected with shRNA^Scramble^ or shRNA^KCTD5^ #1 encoding plasmids. Cells were serum-starved for 4 h and then were treated with 1 μM ionomycin in a Ca^2+^-free (0 Ca^2+^) or a Ca^2+^-containing (2 mM Ca^2+^) Krebs medium for 20 min. Focal adhesions were labeled with mouse mAb anti-vinculin (red). Focal adhesions of EGFP positive cells were analyzed. Scale bar = 25 μm. (**E**) Graph representing the Average of focal adhesion area per cell (µm^2^) for each condition (mean ± SD; *n* = 4; Two-way ANOVA followed by Sidak’s multiple comparisons test, ** *p* < 0.01, **** *p* < 0.0001). (**F**) Graph representing the Number of focal adhesions per cell for each condition (mean ± SD; *n* = 4; Two-way ANOVA followed by Sidak’s multiple comparisons test, *** *p* < 0.001).

**Figure 7 cells-09-02273-f007:**
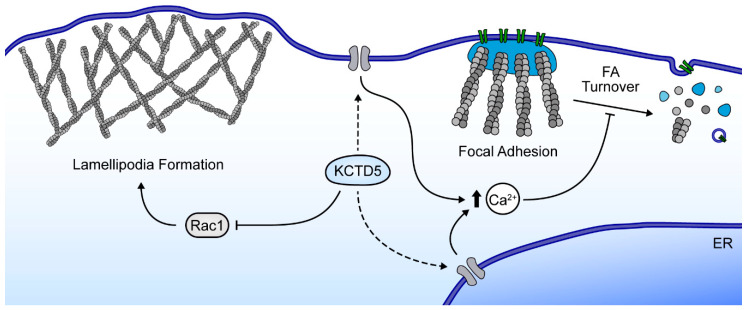
Proposed model. KCTD5 regulates cell migration by two paths: Ca^2+^ signaling and Rac1 activity. KCTD5 promotes the serum-induced Ca^2+^ entry and/or Ca^2+^ release from intracellular storage, leading to a higher global Ca^2+^ signal. This global Ca^2+^ rise inhibits the focal adhesion turnover. On the other hand, KCTD5 inhibits the Rac1 activity, leading to a decreased lamellipodia formation. All these effects suggest that KCTD5 plays a role as a negative modulator for B16-F10 cells migration.

**Table 1 cells-09-02273-t001:** List of antibodies.

Antibody	Isotype	Dilution	Final µg/mL	Source	Type	Catalog#	RRID	Purifi- Cation
Mouse anti-KCTD5	IgG2b	1:1000 (IB)	1	Origene	mAb	TA501035	AB_11140321	Asc
Mouse anti-tubulin	IgG1	1:5000 (IB)	1	Sigma-Aldrich	mAb	T5168	AB_477579	Asc
Mouse anti-GFP	IgG2a	1:2000 (IB)	0.1	Santa Cruz Biotechnology, Inc.	mAb	sc-9996	AB_627695	NA
Mouse anti-vinculin	IgG1	1:200 (IF)	NA	Sigma-Aldrich	mAb	V4505	AB_477617	Asc
Mouse anti-c-Myc	IgG1	1:200 (IF)	10	Sigma-Aldrich	mAb	M4439	AB_439694	AP
Mouse anti-c-Myc	IgG1	1:2000 (IB)	1	Sigma-Aldrich	mAb	M4439	AB_439694	AP
Mouse anti-Rac1	IgG	1:500 (IB)	1	Cytoskeleton, Inc.	mAb	ARC03	AB_2721173	NA
Alexa Fluor 555 conjugated goat anti-Mouse IgG1	IgG	1:1000	2	Thermo Fisher Scientific	pAb	A-21127	AB_2535769	AP
Alexa Fluor 647 conjugated goat anti-Mouse IgG1	IgG	1:1000	2	Thermo Fisher Scientific	pAb	A-21240	AB_2535809	AP

pAb: polyclonal antibody; mAb: monoclonal antibody; AP: affinity purified; Asc: purified from mouse ascites fluids by affinity chromatography; IF: Immunofluorescence; IB: Immunoblot; NA: Not Available.

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
