# Peer review of "K+ Channel Tetramerization Domain 5 (KCTD5) Protein Regulates Cell Migration, Focal Adhesion Dynamics and Spreading through Modulation of Ca2+ Signaling and Rac1 Activity"

_cells, 2020, doi:10.3390/cells9102273_

Round 1
Reviewer 1 Report
In this manuscript, the authors propose the regulation of cell migration by KCTD5, via Ca signalling and Rac1 activity regulation. This is a well design, straight forward work with interesting results. However, I have some comments and questions that might help improve the manuscript. In particular, I think it would be highly advisable to perform some of the experiments in another cell line to strengthen the conclusion.
Major points:
- It is reassuring to see the results reproduced by 2 different suppression methods and to observe the rescue of the phenotype. However, it would be nice to see the results reproduced in another cell line, perhaps another melanoma cell line, to discard cell specific effects. At least some of the functional migration or adhesion experiments with a suppression system should be included.
- Most images are clear but images showing more than one cell and close ups would desirable.
- On Figure 3, the results showing an abrogation of Rac1 driven cell spreading upon KCTD5 KO suggest that KCTD acts downstream of Rac1 to control spreading. The interpretation of the authors is the opposite. Can they further explain?
- On Figure 4H-I, the number of FA assembly analyzed is very small compared with the number of FA disassembly. Can this account for the lack of statistical signification?
- Movies of FA should be included with the submission.
- What is the connection between KCTD function and Rac1 activity? Is there another Rho GTPase involved? The degradation of some Rho GTPases can affect Rac1 activity. Have the levels of other Rho GTPases been checked?
Minor points:
- I understand from the images that shRNA plasmid encodes GFP as well. This fact is not correctly explained in the methods section.
- A definition of lamellipodial cells would be informative.
- On Figure 4, panel A, the cell for the shRNA#2 is not a good example, as it does not have lamellipodia.
Reviewer 2 Report
It is a very interesting basic research study, in which the authors, using a melanoma cell model, elegantly describe how K+ channel tetramerization domain 5 protein controls B16-F10 cells and B16-F10-derived cell lines migration and actin cytoskeleton rearrangement, measuring focal adhesion dymanics and spreading. Canales et al reveal that KCTD5 protein requires the action of Rac1 and the intracellular calcium mobilization to regulate these processes.
It is a very well structured and written manuscript, easy to read and follow and I am convinced that it will be of interest to the readership of Cells journal.
As minor points that should be corrected:
Figure 3D should be explained in the text and in line 73 where GCPR is written it must be GPCR.
Reviewer 3 Report
- In figure 2 phase contrast images should be included.
- In figure 2, why in some pictures only one cell is displayed. More cells are required.
- In figure 2 arrows should be included.
- These cells move randomly or they move towards a gradient? Please discuss.
- Figure 3 E scale bar is missing.
- Why there are different cells shape, for instance comparing Vinculin Figure 2 A, D and 4 A, D.
- Concerning representative images showing tracked focal adhesions why background is so different between figures. It would be better if figures would be bigger. It was hard to follow.
Round 2
Reviewer 1 Report
Thanks for your reply and the additions to the manuscript. I believe the manuscript would improve if you add some of the discussion on your reply to the manuscript (such as the potential mechanisms involving other Rho GTPases).
Also, the new results provided somehow confirm the ones obtained with the melanoma cell line, but are performed on a breast cancer cell line (actually the legend is mislabeled). The part of the discussion referring specifically to melanoma should be revised. It is unfortunate not to have more complete data to include, but I understand the constraints that the current situations impose. However, I would be inclined to add these results as supplementary data.
Author Response
Reviewer 1:
Comments and Suggestions for Authors
Thanks for your reply and the additions to the manuscript. I believe the manuscript would improve if you add some of the discussion on your reply to the manuscript (such as the potential mechanisms involving other Rho GTPases).
We thank the reviewer for the careful reading of our manuscript. We have included the suggested section (please refer to page 17, lines 536-542, and page 18, lines 587-597)
Also, the new results provided somehow confirm the ones obtained with the melanoma cell line, but are performed on a breast cancer cell line (actually the legend is mislabeled). The part of the discussion referring specifically to melanoma should be revised. It is unfortunate not to have more complete data to include, but I understand the constraints that the current situations impose. However, I would be inclined to add these results as supplementary data.
We have included the suggested figure (please see Figure S1).